# Morphology-Dependent Interactions between α-Synuclein Monomers and Fibrils

**DOI:** 10.3390/ijms24065191

**Published:** 2023-03-08

**Authors:** Tinna Pálmadóttir, Christopher A. Waudby, Katja Bernfur, John Christodoulou, Sara Linse, Anders Malmendal

**Affiliations:** 1Biochemistry and Structural Biology, Lund University, P.O. Box 124, SE-221 00 Lund, Sweden; katja.bernfur@biochemistry.lu.se (K.B.); amalm@ruc.dk (A.M.); 2Institute of Structural and Molecular Biology, University College and Birkbeck College, London WC1E 7HX, UK; c.waudby@ucl.ac.uk (C.A.W.); j.christodoulou@ucl.ac.uk (J.C.); 3School of Pharmacy, University College London, London WC1N 1AX, UK; 4Department of Science and Environment, Roskilde University, P.O. Box 260, DK-4000 Roskilde, Denmark

**Keywords:** morphology, polymorphic, monomorphic, self-assembly, NMR spectroscopy, aggregation, stability

## Abstract

Amyloid fibrils may adopt different morphologies depending on the solution conditions and the protein sequence. Here, we show that two chemically identical but morphologically distinct α-synuclein fibrils can form under identical conditions. This was observed by nuclear magnetic resonance (NMR), circular dichroism (CD), and fluorescence spectroscopy, as well as by cryo-transmission electron microscopy (cryo-TEM). The results show different surface properties of the two morphologies, A and B. NMR measurements show that monomers interact differently with the different fibril surfaces. Only a small part of the N-terminus of the monomer interacts with the fibril surface of morphology A, compared to a larger part of the monomer for morphology B. Differences in ThT binding seen by fluorescence titrations, and mesoscopic structures seen by cryo-TEM, support the conclusion of the two morphologies having different surface properties. Fibrils of morphology B were found to have lower solubility than A. This indicates that fibrils of morphology B are thermodynamically more stable, implying a chemical potential of fibrils of morphology B that is lower than that of morphology A. Consequently, at prolonged incubation time, fibrils of morphology B remained B, while an initially monomorphic sample of morphology A gradually transformed to B.

## 1. Introduction

Protein folding is a process by which proteins reach their three-dimensional structure, their native functional form [1]. However, some proteins do not form a defined three-dimensional structure and remain largely unfolded and disordered; these proteins are referred to as intrinsically disordered proteins (IDPs) and are often involved in cellular signaling and regulation [2,3]. In many cases, folding–unfolding processes are fully reversible and typically occur on a µs to ms timescale. Proteins can also misfold and form aggregates that constitute a separate phase [2,4]. The folding and misfolding of proteins can be investigated at equilibrium or as a function of time using various techniques including optical spectroscopy, NMR spectroscopy, and stopped-flow techniques.

A number of amyloidosis diseases, such as Parkinson’s disease, Alzheimer’s disease, Huntington’s disease, and type II diabetes, involve the misfolding of proteins and the formation of amyloid fibrils. These amyloid fibrils are elongated, unbranched, highly stable and ordered aggregates, which are characterized by a cross-β X-ray diffraction pattern [2,4,5,6,7]. The fibrils are composed of assembled monomers with β-strands arranged perpendicular to the fibril axis, forming hydrogen-bonded β-sheets between monomers parallel to the fibril axis, giving rise to a strong β-sheet signal in a circular dichroism spectrum (CD) [4,5,8]. Amyloid fibrils generally consist of thousands of monomers, with a diameter of up to a few nanometers (5–20 nm) and a length of several micrometers. The mature fibrils usually consist of two or more protofilaments that twist around each other [2,8,9]. Structural studies of amyloid fibrils using cryo-electron microscopy (cryo-EM), atomic force microscopy (AFM), and solid-state NMR (ssNMR) have contributed detailed knowledge about the arrangement of the monomers within the fibrils and the arrangement of the β-sheets formed between different monomers along the fibril axis, as well as contacts between specific side-chains and moieties [9].

It has been found that identical monomers can have different intra- and intermolecular interactions, giving rise to supramolecular structures with different fibril morphology (Figure 1A). This can be affected by differences in conditions, such as temperature, pH, salt concentration, and buffer composition [2,9,10]. Furthermore, it has also been shown that identical monomers can form supramolecular structures with different morphologies coexisting in the same sample; such samples are termed polymorphic [9,11,12]. Different morphologies of the same peptide/protein may be related to differences in packing of the protofilaments into mature fibrils [11,13,14,15]. Differences in structures on the atomic level (packing of the monomers into protofilaments) can also cause differences in mesoscopic structure (on micrometer to nanometer scale), resulting in, for instance, different fibril length and protofilament packing, node-to-node distance, and the number of protofilaments twisting around each other [2,9,10]. The morphology of the fibrils can vary, e.g., straight, twisted, or helical [2,9]. Additionally, different morphs will have different surface properties (Figure 1A), which will be further addressed in this paper. This shows that despite all amyloid fibrils having in common being elongated structures with ordered cross-β-sheet motif, they can still have different local packing, different morphologies, and thus different properties even when formed from the same type of monomer. Polymorphism would, however, only exist at equilibrium if all morphs have exactly the same stability.

Free energy landscapes and protein folding funnels are often used as conceptual representations of amyloid formation [2,4]. The broad opening of the funnel at the top represents the many possible conformations for an unfolded protein, with high conformational entropy and high free energy. The bottom of the funnel contains fewer and narrower wells for discrete conformations, representing more stable states with lower free energies [2,16]. One of those wells will include the energy minimum for the native structure of the protein. However, there are also other energy minima possible, which involve the formation of partially folded states, oligomers, and misfolded aggregates, such as amorphous aggregates and amyloid fibrils [2,17]. The part of the folding funnel representing misfolded proteins can include many closely positioned minima of amyloid fibrils with different morphologies [2,12,18]. Between the minima, there may be high or low energy barriers, which govern the kinetic stability of the different states.

Amyloids are non-covalent assemblies, and their formation is thus reversible. The relative rates of all forward and backward reactions change over time as an equilibrium is being established. For example, elongation by monomer addition is more rapid than monomer dissociation during the formation reaction in an initially supersaturated solution; however, at equilibrium, the rates are balanced so that there will be no net change in the fibril or monomer concentration. If the system is diluted, the backward reaction rate will be highest and there will be a net dissolution until a new equilibrium is established, after which the rates are again balanced. In cases of co-existing morphologies of different stability, the reversibility will ensure a net reaction flow towards the more stable morph, although this may be an overall slow process.

Above the solubility limit, the free monomer concentration at equilibrium is constant and independent of the amount of the fibrillar aggregates. Solubility is thus defined as the concentration in the fluid phase in equilibrium with a solid phase (in this case, fibrils). As a consequence of the second law of thermodynamics, the chemical potential of a substance, e.g., an amyloid peptide, will be the same in every phase of the system at equilibrium [19,20]. Thus, different morphs can only co-exist at equilibrium if the chemical potential of monomers in solution is identical to that in monomers in every morph. While this is unlikely in most cases, an observed polymorphism could mean that the system is not yet at equilibrium and that the kinetic barriers separating monomers and the various morphs are similar, while the barriers between morphs are high. Moreover, the apparent solubility will be lower the more stable the morph, and at infinite time, such systems should rearrange to monomorphic. Most likely, such rearrangement involves monomer dissociation from fibril ends and regrowth. In the case of in vivo-derived samples, there may be monomer heterogeneity, e.g., variation in posttranslational modification or length variants, as in the case of amyloid β peptide, meaning that there is more than one monomer type and these monomers may have different solubility and may assemble as separate morphs [21].

α-synuclein, the focus of this study, is an intrinsically disordered protein, and its formation into amyloid fibrils is a hallmark of Parkinson’s disease [22]. A distinct characteristic of α-synuclein is its segregated primary structure with an N-terminal amphipathic region, a central hydrophobic NAC (non-amyloid-β component) region that usually forms the fibril core, and a C-terminal acidic tail [23,24,25,26] (Figure 1B,C). The sequence can also be divided into the three regions: the N-terminal tail, the hydrophobic fibril core, and the C-terminal tail, according to how the monomers form into fibrils, as well as the charge distribution within the sequence [23,24,27,28] (Figure 1B). In the monomer, the two termini are slightly attracted by each other, which disfavors interactions between hydrophobic NAC regions in separate monomers [25,26,29]. The 40-amino-acid-residues-long C-terminal tail protrudes from the surface of fibrils. The tails, with 15 acidic groups each, provide a “fuzzy coat” to which the N-termini from α-synuclein monomers may adsorb [27,30,31]. This mode of interaction would disconnect the N- and C-termini of the individual monomer, resulting in a more extended monomer conformation and facilitating the interaction between NAC regions of monomers at the surface, potentially favoring nucleation (Figure 1D) [31]. The monomer fibril–interaction is reminiscent of the more thoroughly studied interaction between α-synuclein monomers and C-terminal tails decorating negatively charged phospholipid vesicles with adsorbed α-synuclein [32,33,34,35,36,37].

Previous studies of α-synuclein have revealed structures of different morphologies in different samples and coexisting within the same sample [11,14,38]. However, in this study, we obtained fibrils of two different morphologies in different samples formed under the exact same conditions, not coexisting but originating from the same solution. The two morphs show different solubility, CD spectrum, Thioflavin-T interactions, and ultrastructure by cryo-EM. The interaction of the monomer with the different fibrillar surface was studied using solution state NMR. The data reveal two entirely different modes of interaction between monomers and fibrils depending on the fibril morphology.

**Figure 1 ijms-24-05191-f001:**
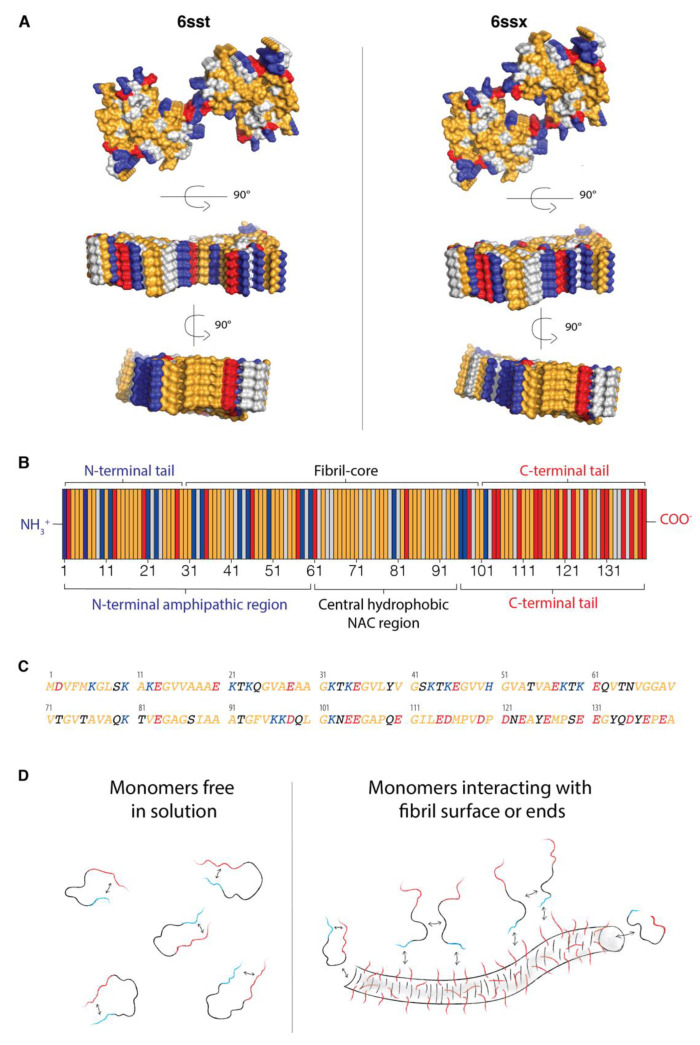
Structure, surface properties, sequence, and interaction modes of α-synuclein monomers and fibrils. (**A**) Examples of structures of α-synuclein fibrils with two different morphologies (PDB: 6ssx and 6sst) containing two protofilaments each. Both structures are solved for residues Gly14-Gly25 and Gly36-Lys96; the N- and C-terminal tails are not part of the ordered structure. The figures are generated in Pymol. Each structure is shown from three different angles. Both structures are shown from the same angles for comparison of the surface properties of the two morphologies. Residues are colored according to their properties: acidic (red), basic (blue), hydrophobic (yellow), and non-charged polar residues (light gray) (**B**) Distribution of acidic (red), basic (blue), hydrophobic (yellow), and non-charged polar residues (light gray). At the top, the sequence is divided into three regions according to how the monomer folds into the fibril structure (2NOA.pdb): the N-terminal tail (residues 1–28), the fibril core (residues 29–100), and the C-terminal tail (residues 101–140). At the bottom, the sequence is divided into the three regions according to sequence properties: the N-terminal amphipathic region (residues 1–60), the central hydrophobic NAC region (residues 61–95), and the acidic C-terminal tail (residues 96–140). (**C**) The sequence of α-synuclein colored according to acidic residues (red), basic residues (blue), hydrophobic (yellow), and non-charged polar residues (black). (**D**) A simplified illustration of α-synuclein monomers free in solution versus interacting with fibrils’ surface and fibril-ends based on the results in references [25,26,29]. The N-terminal tail and the C-terminal tail are shown in red and blue, respectively. The fibril is shown with the acidic C-terminal tails (red) extending from the fibril core, providing an acidic “fuzzy coat” that the N-termini from α-synuclein monomers (blue) may interact with [27,30,31]. The interaction between the N-termini of the monomers and the fibrils’ surface may disfavor the interaction between the N- and C-termini of the monomers, resulting in a more extended monomer conformation. This might facilitate the interaction between the NAC regions of different monomers at the fibril surface, favoring nucleation [31]. The α-synuclein monomers can also interact with the fibril ends. Arrows represent possible interactions between N- and C-termini of the monomers, between N-termini of monomers and “fuzzy coat” of the fibrils, between a monomer and fibril end, and between central hydrophobic NAC region of different monomers that interact with the fibril surface.

## 2. Results

α-synuclein fibrils formed at pH 7.0 (10 mM Tris, 0.02% NaN_3_, 5% D_2_O) were found to adopt two different morphologies, here termed morphology A and morphology B. The fibril samples consisted of fibrils of either morphology A or morphology B, and there was no evidence of co-existence. This behavior was observed by four different techniques: CD, NMR, and fluorescence spectroscopy, as well as cryo-TEM, whereby the two morphologies showed differences in: CD spectra, NMR signal intensities from monomers (solubility), NMR signal intensity profiles within monomers (interaction of monomers with the fibril surface), ^15^N transverse relaxation rates (interaction of monomers with the fibril surface), binding to ThT, and mesoscopic structures.

### 2.1. Fibrils of Two Different Morphologies Examined by CD Spectroscopy, ^15^N−^1^H Signal Intensities, and Cryo-TEM

Figure 2 shows data obtained from two different fibrillar samples, one consisting of fibrils with morphology A and the other one with morphology B. It is important to emphasize that the samples presented in Figure 2 originated from exactly the same monomeric solution. The samples were prepared as explained in Section 4.3, and before forming the fibrils (37 °C with stirring), an equal amount of the monomeric sample was split into two separate low-binding tubes containing the same type of stir bar. After two days of incubation, both samples gave a β-sheet signal. However, they were shifted relative to each other. Morphology A gave a CD spectrum with a negative peak at 216 nm and a positive peak at 192 nm, while morphology B gave a CD spectrum that was shifted to a higher wavelength, with a negative peak at 220 nm and a positive peak at 198 nm (Figure 2A). CD spectra were obtained for 17 independent α-synuclein fibril samples formed at the same condition. These spectra can be divided into two groups, one similar to morphology A and the other similar to morphology B (see Appendix A).

Different behavior between the two samples was also observed by NMR spectroscopy. The ^15^N−^1^H signal intensities of residue A140 of both samples (morphologies A and B) were compared to that of a monomeric sample in order to calculate the concentration of free monomers remaining in the two samples. Assuming that the C-terminal tail of the monomer is not affected by any transient interactions (as suggested by the transverse relaxation data, see below, Section 2.2), the fibril sample consisting of fibrils of morphology A showed a higher solubility, with an intensity of A140 relative to the purely monomeric sample of 0.38 ± 0.04, corresponding to a monomeric concentration of 27.7 ± 3.0 µM. The sample with fibrils of morphology B showed a lower solubility, with an intensity of A140 relative to the purely monomeric sample of 0.08 ± 0.02, corresponding to a monomeric concentration of 6.0 ± 1.2 µM (Figure 2B).

Cryo-TEM was used to evaluate ultrastructural differences between the two fibril samples (morphologies A and B) and a clear difference was seen (Figure 2C–F). The fibrils of morphology A were thinner compared to the fibrils of morphology B, with a diameter of 11.1 ± 1.3 nm and 18.0 ± 4.9, respectively. The fibrils of morphology A seemed to be formed from two protofilaments twisting around each other while fibrils of morphology B were made from a higher number of protofilaments. Fibrils of morphology A were more twisted with a measurable node-to-node distance of 91 nm ± 8 nm. The fibrils of morphology B were straight and ribbon-like, with several protofilaments aligned next to each other; the number varied between individual fibrils, reflected in the larger standard deviation of the diameter for morphology B. The node-to-node distance was much longer and not measurable for morphology B. The sample containing fibrils of morphology B also contained a higher number of fibrils compared to the sample of fibrils of morphology A. The greater number of fibrils observed in the sample of morphology B is consistent with the lower monomer solubility as measured by NMR. Assuming that no other NMR invisible oligomers are present, there are 43 µM and 64 µM of monomers making up fibrils of morphologies A and B, respectively, in the NMR samples. As mentioned in Section 4.7, the sample containing fibrils of morphology B had to be diluted 1:1 in the sample buffer due to a higher viscosity that caused troubles in blotting, which might be explained by the higher number of fibrils.

We tested whether the difference in concentration of monomers in equilibrium with the fibrils of the two morphologies (A and B) is the reason for the difference in the CD spectra, and if that could explain the shift of the peaks between the two morphologies. This was achieved by taking CD spectra of a fibril sample mixed with a monomeric sample at different ratios (see Appendix A). CD spectra of a monomeric sample and a fibril sample were also used to calculate a theoretical spectrum containing only fibrils (without soluble monomers) (see Appendix A). The results showed that the shift in the β-sheet signal between the two morphologies cannot be explained by the difference in solubility between the two morphologies.

### 2.2. Variation in ^15^N−^1^H Signal Intensities and Residue-Resolved ^15^N Transverse Relaxation Rates

Samples were prepared the same way as those analyzed in Section 2.1. CD spectra were taken of both samples to verify the secondary structure. The two different samples gave CD spectra similar to the ones shown in Figure 2A, where one of the samples showed a β-sheet signal corresponding to morphology A while the other sample showed a β-sheet signal shifted towards higher wavelengths, corresponding to morphology B (Figure 3A).

^15^N–^1^H signal intensities were measured for pure monomers before fibril formation and for monomers after formation of morphology A or B. The relative intensity of the monomer signals per amino acid residue after the formation of morphology A or B are shown in Figure 3B. For both morphologies there is a variation in intensity over the amino acid sequence due to transient monomer–fibril interactions. In the presence of fibrils of morphology A, it is mostly the N-terminal 16 residues that are affected, and the lowest visible intensities of Met5-Leu8 are attenuated ~30-fold relative to the C-terminal A140. In the presence of fibrils of morphology B, there is a similar attenuation (~35-fold), but a very gradual increase in signal intensity from the N-terminus to around Val40, then a bit of a steeper increase until Lys102, after which the intensity remains relatively flat towards the C-terminus.

If we assume that the C-terminal tail of the monomer is not affected by any transient interactions (as suggested by the transverse relaxation data, see below), we may estimate that 30% of the monomer is free in the presence of morphology A and 10% is free in the presence of morphology B, resulting in free monomer concentrations of 21 and 7 µM, respectively. The free monomer concentration is consistent with the results shown in Figure 2B.

^15^N transverse relaxation rates depend on the pico- to nanosecond mobility of the amide ^15^N–^1^H bond vectors. In short, a higher rate corresponds to a more dynamic bond vector. For ^15^N nuclei that experience conformational exchange between states with differing chemical shifts, the corresponding transverse relaxation rates may contain an additional exchange contribution, Δ*R*_2_. If a ^15^N nucleus experiences chemical shift changes of the same order of magnitude as the exchange rate, they are in the so-called intermediate exchange regime (2π ν_0_ Δδ ≈ *k*_ex_), where ν_0_ is the spectrometer frequency for ^15^N (91.2 MHz in this case), Δδ is the ^15^N chemical shift difference between the two states, and *k*_ex_ is the exchange rate. Here, 2π ν_0_ Δδ is on the order of 100 s^−1^. These nuclei get an exchange contribution, Δ*R*_2,_ to the transverse relaxation rates. If the exchange rate is on the slower side of this limit (slow on the chemical shift timescale), Δ*R*_2_ will only depend on *k*_ex_, while if it is faster (fast on the chemical shift timescale), Δ*R*_2_ will depend on both *k*_ex_ and Δδ. For fast exchange, the measured chemical shift will be a weighted average of the shifts in the two states, while the position of the signals from the two states will be largely unaffected in slow exchange.

For slow exchange with a large object, such as a fibril, the additional exchange contribution is caused by lifetime broadening due to the unidirectional conversion of monomer to an NMR invisible fibril bound state. Here, the Δ*R*_2_ gives an estimate of the apparent first order association rate constant, *k*_on_, for the process. Variation in Δ*R*_2_ along the amino acid sequence suggests a more complicated binding model [39,40].

^15^N transverse relaxation rates were measured for pure monomer before fibril formation and for monomers after the formation of morphology A or B (Figure 3C). In the presence of fibrils of morphology A, the relaxation rates for some of the first 10 residues are higher than for the rest of the protein and much higher than for the corresponding residues in the pure monomer sample, with Δ*R*_2_ of between 3 and 10 s^−1^. Starting at residue 11 (where Δ*R*_2_ is about 2 s^−1^), the relaxation rates get gradually closer and closer to those of the pure monomer until the C-terminal Ala140 (where Δ*R*_2_ is about 0.2 s^−1^).

If we assume that the exchange contributions are due to exchange with the fibril, as has been established for acetylated α-synuclein, this means that the different parts of the monomer interact in different ways with the fibril, corroborating the conclusions made above based on the ^15^N−^1^H signal intensities.

In the presence of fibrils of morphology B, the signal intensities are so low in the N-terminal region that we can only measure relaxation rates from V66 and onwards. Here, a striking feature is that the rates in the presence of fibril are consistently lower than for the pure monomer from E104 and onwards with a peak in rates between V118 and E137. This either indicates that this part of the protein is more flexible in the presence of fibrils of morphology B or that there is some exchange contribution in the monomer that is cancelled in the presence of fibrils of this morphology. Interestingly, many of the residues from A124 and onwards also show ^15^N chemical shift perturbations (Appendix A). All of this suggests that this exchange process is faster. The more flexible nature and chemical shift perturbations for the C-terminal residues may be due to the lack of interaction with the N-terminus when bound to or in the vicinity of the fibril. It is, however, not obvious exactly how this process is related to the interaction with the fibril.

### 2.3. Replication of Morphologies A and B and Evolution with Time

#### 2.3.1. Replication of Morphologies A and B Using Seeds

The replication of fibrils of morphologies A and B was tested by adding 1% seeds of either morphology A or B to freshly prepared monomer solutions (Section 4.4.2). All samples were made from the same monomeric preparation and samples were incubated at 37 °C with stirring throughout the whole experiment. After one day of incubation, samples containing seeds of morphologies A and B displayed separate β-sheet signals, i.e., the sample seeded with fibrils of morphology A had a CD spectrum corresponding to morphology A and the sample seeded with fibrils of morphology B had a CD spectrum corresponding to morphology B (Figure 4A,B, red). This showed that the fibril morphology can be successfully replicated by seeding monomer samples.

#### 2.3.2. Evolution of Morphologies A and B with Increased Incubation Time

CD spectra were taken as incubation went on (after 1 day, 2 days, 3 days, and 6 days) to detect any change in secondary structural content of the samples (Figure 4 and Appendix A). While the CD spectra of the samples containing morphology B were largely unaltered, it was found that the CD spectra of the samples initially containing morphology A became more similar to the CD spectrum of the sample containing morphology B (Figure 4), with a negative peak around 220 nm.

Furthermore, aliquots of the samples were withdrawn at each time point, centrifuged, and the supernatant was analyzed by SDS-PAGE to compare the monomer concentration in the fibril samples to that of a pure monomeric sample (Figure 5). After one day of incubation, the sample supplemented with seeds of morphology A had a higher concentration of monomers than the sample supplemented with fibrils of morphology B, indicating a higher solubility. This is consistent with the previous results, where morphology A showed higher solubility than morphology B when examined with NMR spectroscopy and cryo-TEM (Figure 2 and Figure 3).

From the SDS-PAGE (Figure 5) and mass spectrometry with isotope standard (Appendix A), it can also be seen that the monomer concentration in the sample seeded with fibrils of morphology A decreases with increasing incubation time. Over time, the two types of samples seem to equilibrate to the same free monomer concentration (Appendix A). Fibrils of morphology A are thus associated with a higher apparent solubility than fibrils of morphology B. This is consistent with the time-dependent change in CD spectrum presented in Figure 4, indicating that over time, samples seeded with fibrils of morphology A are substituted by fibrils of morphology B. This indicates that fibrils of morphology B are energetically more stable than those of morphology A (see discussion).

The bands appearing around 30 kDa correspond to dimers as centrifugation does not successfully separate monomers from dimers. The intensity of the band at 30 kDa is largely proportional to the band at 15 kDa, and thus more visible at the higher monomer concentrations in the presence of morphology A. The oligomeric distribution in the presence of morphologies A and B may be further investigated by methods, such as photo-induced cross-linking of unmodified proteins (PICUP).

### 2.4. Binding of Thioflavin-T to Fibrils of Morphologies A and B

The binding of ThT to amyloid fibrils restricts the rotation around the carbon–carbon bond, resulting in increased fluorescence. This makes ThT a useful probe to study amyloid fibrils [41,42,43,44]. Amyloid fibrils of different morphologies have different surface properties, affecting the flexibility around the carbon–carbon bond of the bound ThT and thus the fluorescence intensity. The interaction between ThT and the fibrils, and the resulting fluorescence intensity, has previously been shown to differ between α-synuclein fibrils of different morphologies [45,46]. Here, we investigated the surface properties of morphologies A and B by measuring the fluorescence emission spectra of samples containing either fibrils of morphology A or B titrated with ThT (see Section 4.6) (Figure 6A,B and Appendix A). CD spectra were taken of the samples used for the experiment in order to verify the morphology present in each sample (Figure 6C). The samples were titrated with ThT solution until the maximum intensity of the observed fluorescence spectrum started to decrease. Figure 6A compares the ThT emission spectra with the highest fluorescence intensity. The maximum fluorescence intensity differed between the two morphologies, where the maximum ThT fluorescence intensity for the two replicates of morphology A was more than two times higher than that of the two replicates of morphology B (Figure 6A,B). The concentration of ThT needed to reach the maximum fluorescence also differed between morphologies A and B (Figure 6B). This study found that 35 µM and 17 µM of ThT was needed to reach the maximum fluorescence of morphologies A and B, respectively. The results indicate that the binding of ThT to the two morphologies is different, suggesting a difference in surface character between the two morphologies.

Additionally, the relative concentration of monomer in the samples was compared using SDS-PAGE to estimate the amount of monomer present in the supernatant after centrifugation of the samples. (Figure 6D). The amount of monomer in the samples containing morphology A is higher than in the samples containing morphology B, which is consistent with results obtained by measuring the fraction of free monomers with NMR spectroscopy by calculating the relative intensity of A140 in the fibrillar samples to the intensity of a monomeric sample (Section 2.1 and Section 2.2). The higher ThT fluorescence of the samples containing morphology A can, therefore, not be due to a higher number of fibrils in the samples, as the samples of morphology A contain fibrils at lower concentration than the samples of morphology B.

The supernatant of the different samples after centrifugation was also analyzed by MALDI-TOF Mass Spectrometry of both intact (Figure 6E) and trypsin-digested samples. This was performed in order to check if the monomers in supernatants of both morphologies were chemically identical after incubation and that no modifications had occurred. The data confirmed that all samples contained full-length α-synuclein without any detectable modifications.

## 3. Discussion

### 3.1. The Formation of Morphologies A and B

A significant polymorphism has been reported for α-synuclein fibrils formed under different solution conditions [11,14]. However, at constant solution condition, the second law of thermodynamics implies that each molecule must have the same chemical potential in every phase, irrespective of how many phases are present in the system. Therefore, polymorphism can only exist at equilibrium if two or more morphologies have exactly the same stability. If not, an observed polymorphism is a consequence of kinetics and high energy barriers between the minima representing separate morphologies (Figure 7A). If polymorphism initially arises, only the most stable morphology will persist over time.

The current data show a clear example of an initial polymorphism being a consequence of kinetics rather than thermodynamics. We observe two different morphologies after fibril growth from chemically identical samples, arbitrarily named A and B. The NMR and SDS-PAGE data (Figure 1, Figure 3, Figure 5 and Appendix A) reveal a lower monomer concentration in the presence of fibrils of morphology B compared to A, i.e., a lower apparent solubility (Figure 7B,C). Fibrils of morphology B are thus thermodynamically more stable than those of morphology A (Figure 7A). The gradual transformation over time from morphology A to B (Figure 4 and Figure 5) gives further evidence of a higher stability of fibrils of morphology B and that the end state is a consequence of the thermodynamics. The ability to recreate the less stable A form using seed fibrils of morphology A is a consequence of very high energy barriers for primary nucleation, making it a very rare event, and the lower barriers for elongation and secondary nucleation of fibrils of an existing morphology relative to primary nucleation of fibrils of the more stable morphology. Still, morphology A is only kinetically stable, and the thermodynamically more stable morphology B takes over the sample in the end. The high effective energy barriers between morphologies A and B (Figure 7A) and the very slow primary nucleation process is likely an explanation for why a sample containing morphology A can be monomorphic for a while, before the sample becomes polymorphic (containing both morphologies A and B) and subsequently becomes monomorphic again, consisting only of morphology B.

The NMR studies reveal a significant exchange between monomers that are free in solution and monomers that are associated to the surface of the fibrils. However, we do not from our data have any direct information about the exchange between free monomers and those that are part of the fibril. Still, because of the reversibility of non-covalent fibril formation and low likelihood of collective structural conversion of a formed fibril, the most straightforward mechanism for conversion of a sample towards fibrils of the most stable morphology would be monomer dissociation from fibril ends and nucleation and growth of the more stable form (Figure 7A).

### 3.2. Monomer–Fibril Interaction Differs between Morphologies A and B

NMR was used to investigate the interaction of α-synuclein monomers with fibrils of morphologies A and B. The sequence dependence of the ^15^N−^1^H signal intensities of the soluble monomers was different in the presence of morphologies A and B, indicating that soluble monomers interact differently with the fibril surface of two different morphologies (Figure 3B).

In the presence of fibrils of morphology A, the first 16 residues of the N-terminus of the monomers are mostly affected, and the lowest visible intensities Met5-Leu8 are attenuated ~30-fold relative to the C-terminal A140 (Figure 3B). The observed sequence dependence of the ^15^N−^1^H signal intensities in the presence of morphology A is very similar to what has been described earlier for acetylated α-synuclein, indicating transient interactions between the N-terminus of the monomers and the C-terminus extending from the fibril surface [30,31]. Specifically, there are interactions between the positive charges within the N-terminus of the monomers (Lys6, Lys10, and Lys12) and the negative charges on the C-terminal tails of the fibrils, as well as π-π interactions between Phe4 within the N-termini of the monomers and the aromatic residues of the C-terminal tails of the fibrils [31].

The changes in ^15^N transverse relaxation rates suggest that in the presence of fibrils of morphology A, the first 10 residues interact in another way than the rest of the protein (Figure 3C), in agreement with the main monomer–fibril interaction occurring between the N-terminus of monomers and the fibril surface (Figure 8A). The uniformity of the deviation from the ^15^N transverse relaxation rates of residue 11–140 from those of pure monomer suggests an interaction between these residues and the fibril surface that is slower than 100 s^−1^.

The sequence dependence of the ^15^N−^1^H signal intensities of the soluble monomers in the presence of morphology B is very different from morphology A (Figure 3B) and from what has been shown for acetylated α-synuclein monomers in the presence of fibrils [30,31]. It is more similar to what has been seen for interactions between α-synuclein monomers and negatively charged phospholipid vesicles where a large part of the protein is affected [32,33,34,35], but it is different in that the C-terminal tail is also affected here. One could imagine an interaction where the N-terminus and maybe also the N-terminal part of the NAC region is the major binding site but that the rest of the molecule also interacts to some degree (Figure 8B). The lower ^15^N transverse relaxation rates and the larger chemical shift differences for the C-terminal tail suggest that this region could be involved in an exchange process that is much faster than 100 s^−1^.

### 3.3. Different Surface Properties of Morphologies A and B

The results obtained from NMR, ThT binding, and cryo-TEM imaging all support the conclusion that the two chemically identical morphologies, A and B, have different surface properties. It has earlier been shown that different parts of the protein are exposed in different α-synuclein morphologies [47].

The NMR data (^15^N−^1^H signal intensities and ^15^N transverse relaxation rates) reveal differences in interaction between the monomers and fibrils of morphologies A and B. In contrast to morphology A, where only the first 10 residues at the N-terminus of the monomer interact with the fibril surface, the results indicate that both N-termini and the central-hydrophobic domain of the monomers interact with the fibril surface of morphology B (Figure 3). These results suggest that the surface of fibril of morphology B is stickier compared to morphology A, possibly due to higher hydrophobicity. This correlates well with the cryo-TEM images, where a higher number of protofilaments seems to align together to form morphology B than A, which also indicates less repulsion or stronger attraction between protofilaments of morphology B than A.

Despite the lower solubility of morphology B (higher number of fibrils), the ThT fluorescence intensity was lower than for A. Binding of ThT to the fibril surface differs between the two morphologies. The affinity of ThT for fibril surface seems lower for A than B. This indicates differences in surface properties between the two morphologies.

### 3.4. Conclusions

The results of the current work show that two chemically identical but morphologically different α-synuclein fibrils, here called A and B, may form in replicates of the same solution, i.e., under identical solution conditions. Distinct monomorphic samples arise initially, composed of fibrils of only morphology A or only morphology B. Over time, all samples converge to morphology B, implying that morphology B is thermodynamically more stable than morphology A. During the conversion period, samples with fibrils of morphology A become temporarily polymorphic, but over time, they become monomorphic again, containing fibrils of morphology B only. Our results can be understood in terms of propagation of the first nucleated morphology in a given solution being kinetically favored. Due to the high kinetic barriers for primary nucleation, the replication through secondary nucleation and elongation of a less stable morphology is favored over the primary nucleation of a more stable morphology.

In line with the higher stability of morphology B fibrils, they display significantly lower solubility than morphology A. The difference in solubility implies that the chemical potential of the monomers in the presence of morphologies A and B is not the same. In accordance with the second law of thermodynamics, the two morphologies cannot co-exist at equilibrium (polymorphic sample). Thus, with prolonged incubation time, fibrils of morphology B take over a sample initially containing morphology A, and the sample becomes monomorphic at infinite time.

There are also clear differences between the morphologies in terms of structure, surface properties, and in the interaction of monomers with the fibril surfaces. A small part of the N-terminus of the monomer interacts with the surface of morphology A, while a larger part of the monomer interacts with the surface of morphology B.

## 4. Materials and Methods

### 4.1. Expression of α-Synuclein

#### 4.1.1. Non-Labelled wild-Type α-Synuclein

*Escherichia coli* (*E. coli*) BL21* pLysS Ca^2+^ competent cells were used for the transformation of a pET-3a-plasmid containing the gene for wild-type α-synuclein, with *E. coli* optimized codons and an ATG start codon corresponding to Met1 (purchased from GenScript, Piscataway, NJ, USA) [27]. First, 30 to 40 µL of the competent cells were mixed with 0.7 µL of plasmid (100 ng/µL) and kept on ice for 30–60 min before it was heated for 45 s at 42 °C and placed on ice for an additional 10 min. The cells were plated and incubated overnight (ON) on LB agar plates containing chloramphenicol (30 µg/mL) and ampicillin (50 µg/mL). Negative control was performed the same way but without the plasmid. Next, the plates were stored for 8 h at 5 °C, followed by a selection of a number of small and well-isolated colonies that were each used for inoculation of a 50 mL ON culture (LB medium made from 10 g NaCl, 10 g Bacto^TM^ Trypton, and 10 g Bacto^TM^ yeast extract per L, 30 µg/mL chloramphenicol, and 50 µg/mL ampicillin) at 37 °C with shaking. The following morning, 5 mL of each ON culture was added to 500 mL of LB medium (30 µg/mL chloramphenicol and 50 µg/mL ampicillin) in 2.5 L baffled flasks and incubated at 37 °C with orbital shaking (125 rpm). When the optical density at 600 nm (OD_600_ nm) reached approximately 0.9–1.0, the cultures were induced with 100 µg/mL isopropyl thio-β-D-galactoside (IPTG). The cells were harvested 4 h after the induction by centrifugation for 12 min at 6000 g and 4 °C using a JA 8.100 rotor. The pellets obtained from a total of 4 L of culture were combined, mixed with 25 mL of water, and frozen at −20 °C. Prior to harvesting, 1 mL samples were taken from each culture for analysis of the expression by SDS-PAGE (Appendix A).

#### 4.1.2. N-Labeled Wild-Type α-Synuclein

Expression of ^15^N-labeled wild-type α-synuclein was performed the same way as explained for non-labeled protein, except that M9 minimal medium was used in the day cultures. After incubation of the LB agar (with chloramphenicol (30 µg/mL) and ampicillin (50 µg/mL)) plates ON and subsequently for 8 h at 5 °C, a well-isolated single colony was transferred to 50 mL of LB medium (with 30 µg/mL chloramphenicol and 50 µg/mL ampicillin). The culture was shaken ON at 125 rpm at 37 °C. The next day, 3 mL of the ON culture was transferred to 50 mL of middle-day culture, consisting of M9 minimal medium, with ^15^NH_4_Cl as the sole nitrogen source, supplemented with 30 µL/mL chloramphenicol and 50 µL/mL ampicillin. The OD_600_ was followed, and when it had reached approximately 0.8, 50 mL of the middle-day culture was transferred to 450 mL of M9 minimal medium supplemented with 30 µL/mL chloramphenicol and 50 µL/mL at 37 °C with shaking at 120 rpm. The cultures were induced with 100 µg/mL IPTG when the OD_600_ had reached approximately 0.8. The cells were harvested 4–5 h after induction by centrifugation for 12 min at 6000× *g* and 4 °C in a JA 8.100 rotor. The pellets obtained from a total of 4 L of culture were combined, mixed with 25 mL of water, and frozen at −20 °C. Before harvesting, 1 mL samples were taken from each culture for testing of the expression by SDS-PAGE (see Appendix A).

### 4.2. Purification of α-Synuclein

#### 4.2.1. Handling of Cell Pellets before Chromatography

The purification was performed in the same way for the non-labeled and labeled α-synuclein. Cell pellets obtained from a total of 8 L of culture were thawed and dissolved in 100–120 mL buffer A (10 mM Tris/HCl, 10 mM EDTA, pH 7.5) and placed on ice. Next, the thawed and dissolved cell pellet was sonicated on ice using pulse sonication (1 s on, 1 s off) until the mixture became homogeneous. Thereafter, the sample was centrifuged at 15,000× *g* and 4 °C for 10 min (JA 25.50 rotor) and the supernatant was collected and poured into an equal volume of boiling buffer A. The temperature of the sample was measured until it reached 85 °C; afterwards, the sample container was immediately placed in ice-water slurry, stirred until it had cooled down, and then centrifuged at 15,000× *g* and 4 °C for 10 min (JA 25.50 rotor). This procedure precipitates and removes most of the *E. coli* proteins.

#### 4.2.2. Two Steps of Ion-Exchange Chromatography

The supernatant from above was subjected to two steps of ion-exchange chromatography. The first step was performed using diethylaminoethyl (DEAE) cellulose. Buffers were always degassed, filtered (hydrophilic polypropylene membrane filters, 0.2 µm, Pall Corporation), and kept cold throughout purification; furthermore, purification was performed in a cold room. The resin was washed two times with degassed milli-Q water and then two times with buffer A, or until the pH of the resin in buffer was around 7.5. The column (3.5 cm) was packed and then equilibrated with 100 mL of buffer A. The sample was loaded slowly (approximately 2 mL/min) onto the column using a pump. Thereafter, the column was washed with a minimum of 100 mL of buffer A. At the end of the washing step, the flow rate was lowered to 1 mL/min. When the flow rate was stable (1 mL/min) the sample was eluted with a 0–0.5 M NaCl gradient in buffer A. The eluted sample was collected in fractions and analyzed with SDS-PAGE. The fractions containing most of the α-synuclein and a minimum of impurities were combined and diluted 1:1 with buffer A and subsequently purified with the second ion-exchange chromatography step, performed as described above but using 60 g of DEAE sephacel resin packed into a column with a diameter of 2.3 cm. The different fractions were first analyzed by measuring the absorbance at 280 nm, and then the fractions with absorbance at 280 nm were analyzed with SDS-PAGE to determine the purity as well as the presence of α-synuclein. The fractions containing α-synuclein and no detectable impurities were pooled and stored at −20 °C in 1 mL aliquots. The concentration of the pooled sample (in the range of 1–3 mg/mL) was measured by absorbance at 280 nm, using an extinction coefficient of ε_280_ = 5800 M^−1^cm^−1^. The correct mass and lack of other species was confirmed using MALDI mass spectrometry (see Appendix A).

### 4.3. Monomer Preparation

First, 1 mL aliquots of α-synuclein were lyophilized. To concentrate the sample, a few lyophilized aliquots were dissolved in a smaller volume of 6 M guanidinium hydrochloride, to a total volume of 1.1 mL. To make sure that the sample was fully dissolved, it was incubated for at least 1 h at RT before it was loaded (1 mL) onto a Superdex 75 Increase 10/300 GL (GE Healthcare) size exclusion column (SEC) using a fast protein liquid chromatography (FPLC) system (Bio-RAD BIOLOgic Duo Flow, Hercules, CA, USA). The sample was eluted at a flow rate of 0.7 mL/min in the experimental buffer (10 mM Tris, 0.02% NaN_3_, pH 7.0). The buffer was freshly prepared, filtered, and degassed before the start of each experiment. The absorbance at 280 nm was used to follow the elution of the monomers. The fractions corresponding to the center of the monomer peak (1.0–1.5 mL) were collected into low-binding tubes (Genuine Axygen Quality) and kept on ice until further use. The concentration of the monomeric sample was determined from the absorbance at 280 nm, using an extinction coefficient of ε_280_ = 5800 M^−1^cm^−1^. The monomeric sample was diluted to 70 µM in 10 mM Tris, 0.02% NaN_3_, pH 7.0. Samples analyzed with NMR spectroscopy contained 5% D_2_O.

### 4.4. Fibril Formation

#### 4.4.1. Fibril Formation with Stirring

Fibrils were formed by incubating 70 µM of freshly purified α-synuclein monomers in a 2 mL low binding-tube (Genuine Axygen Quality) at 37 °C with continuous stirring at 700 rpm using a micro stirring bar (8 × 3 mm, polytetrafluoroethylene (PFTE)-coated, strong Alnico V magnetic core, round smooth surface (distributed by VWR International, made in UK)). The presence of fibrils was confirmed using CD spectroscopy.

#### 4.4.2. Replication of Morphs—Seeded Fibril Formation with Stirring

For replicating the two different morphologies (termed A and B), the monomeric sample was supplemented with 1% seeds (fibrils of morphology A or B) before incubation at 37 °C and stirring at 700 rpm, as above. Seeds were tested with CD-spectroscopy and sonicated with a sonication bath for 1 min prior to use.

### 4.5. Far-UV Circular Dichroism (CD) Spectroscopy

Far-UV spectra were recorded using a Jasco J-815 CD spectrometer between 260 nm and 190 nm at 20 °C using a quartz cuvette with a path length of 0.1 mm, a scanning speed of 50 nm/min, continuous scanning mode, digital integration time per data point (D.I.T) of 8 s, and sensitivity set to standard. The data from three accumulations were averaged.

### 4.6. Binding to ThT

The binding of Thioflavin-T (ThT) to fibrils of different morphologies was investigated by titrating ThT into α-synuclein fibril samples containing fibrils of either morphology A or B. The fibril samples used for the experiment were made from 70 µM monomeric samples that had been seeded with seeds of corresponding morphology (A or B) and incubated for 1 day at 37 °C. The CD spectra of the samples were recorded before the experiment started, to verify the morphology of the samples. Samples were diluted to 10 µM (in 10 mM Tris, 0.02% NaN_3_, pH 7.0) and titrated with 1 mM ThT stock solution (dissolved in water). The samples were excited at 440 nm, and emission spectra were recorded from 450 to 550 nm, with excitation and emission slits set to 4 nm using a Perkin Elmer Luminescence Spectrometer LS-50B (UK). Two replicates of each morphology were titrated. The average intensity between 477 and 482 nm was plotted against the ThT concentration.

### 4.7. Cryogenic Transmission Electron Microscopy (Cryo-TEM)

Two different morphologies of fibrils (morphologies A and B) formed in samples with a total monomeric concertation of 70 µM were analyzed with cryo-TEM. To ensure proper mixing, both samples were carefully pipetted up and down before they were applied to grids and frozen. The samples containing the fibrils of morphology B were diluted 1:1 in buffer prior to freezing of the sample due to higher viscosity that caused troubles in blotting. The specimens for cryo-TEM were prepared in an automatic plunge freezer system (Leica EM GP). The climate chamber temperature was kept at 21 °C, and relative humidity was ≥90% to minimize loss of solution during sample preparation. The specimens were prepared by placing 4 μL solution on glow discharged lacey formvar carbon-coated copper grids (Ted Pella) and blotted with filter paper before being plunged into liquid ethane at –183 °C. This leads to vitrified specimens, avoiding component segmentation and rearrangement and the formation of water crystals, thereby preserving original microstructures. The vitrified specimens were stored under liquid nitrogen until measured. A Fischione Model 2550 cryo transfer tomography holder was used to transfer the specimen into the electron microscope, JEM 2200FS, equipped with an in-column energy filter (Omega filter), which allows for zero-loss imaging. The acceleration voltage was 200 kV and zero-loss images were recorded digitally with a TVIPS F416 camera using SerialEM under low-dose conditions with a 10 eV energy-selecting slit in place.

The node-to-node distance and the diameter was measured using the ruler tool in the program Adobe Photoshop. The node-to-node distance was measured at 28 different positions. The diameter was measured in between the nodes, where the fibrils appeared the broadest. The diameter was measured from 100 different positions in each case.

### 4.8. Sodium Dodecyl-Sulfate Polyacrylamide Gel Electrophoresis (SDS-PAGE) Analysis

The concentrations of monomers present in the fibril samples were analyzed using SDS-PAGE with Novex^TM^ 10–20% Tricine gels (Invitrogen by Thermo Fisher Scientific). Aliquots (35 µL) were taken from the samples at different time points and centrifuged for 10 min at 14.500 rpm. The supernatant was collected (20 µL) and frozen at −20 °C for later analysis. When all samples had been collected, the frozen supernatants were thawed and 7 µL was mixed with 7 µL of loading buffer. To prepare a standard curve, 7 µL of the original monomeric sample (70, 35, and 18 µM) was also mixed with 7 µL of loading buffer. A sample volume of 10 µL was loaded onto each well, as well as 3 µL of PageRuler^TM^ prestained protein ladder (Thermo Fisher Scientific, Waltham, MA, USA). Gels were stained using Instant Blue^TM^ Protein Stain, Expedeon (United States Biological Life Sciences, Salem, MA; USA).

### 4.9. NMR Spectroscopy

All samples for NMR spectroscopy were prepared at a starting monomer concentration of 70 µM in 10 mM Tris, pH 7.0, 0.02% NaN_3_, 5% D_2_O (Section 4.3). Fibrils were formed as described in Section 4.4.1. After preparation of the individual monomeric and fibrillar samples, they were added to NMR tubes and analyzed.

In order to estimate the free monomer concentrations in the samples, the ^15^N−^1^H signal intensity of A140 was measured for pure monomer before fibril formation and for monomers in the presence of either of the two morphologies. Spectra were recorded at 298 K using an Agilent VNMRS DirectDrive spectrometer operating at a ^1^H frequency of 499.9 MHz and equipped with a 5 mm room temperature probe. ^15^N−^1^H signal intensities were measured using the gNhsqc pulse sequence. A total of 609 × 128 data points were collected in 48 scans for all samples.

For a more in-depth characterization of the monomer–fibril interactions, ^15^N−^1^H signal intensities and ^15^N transverse relaxation rates were measured for pure monomer before fibril formation and for monomers in the presence of either of the two morphologies. Spectra were recorded at 298 K using a Bruker Avance III HD 900 spectrometer (Bruker Biospin, Rheinstetten, Germany) operating at a ^1^H frequency of 899.8 MHz and equipped with a 5 mm cold probe. ^15^N−^1^H signal intensities were measured using the sfhmqcf3gpph pulse sequence. A total of 3072 × 256 data points were collected. ^15^N transverse relaxation rates were measured using a modified version of the hsqct2etf3gpsi3d pulse sequence. Spectra with relaxation delays of 0, 29, 56, 84, 112, 169, 225, 281, 337, 393, and 449 ms were recorded in an interleaved fashion. A total of 11 × 3072 × 160 data points were collected. Since the concentration of free monomer was different in the three types of samples studied, a different number of scans were collected. For pure monomer we collected 2 scans for intensities and 8 scans for relaxation rates; for morphology A we collected 2 scans for intensities and 2 times 8 scans for relaxation rates; and for morphology B we collected 256 scans for intensities and 3 times 32 scans for relaxation rates.

Spectra were processed using the topspin software (Bruker Biospin, Rheinstetten, Germany), and intensities and relaxation rates were measured in PINT [48,49].

### 4.10. Mass Spectrometry

#### 4.10.1. MALDI Mass Spectrometry for Intact Weight Analysis

Protein samples were diluted 1:2 with 0.1% TFA before 1 µL of the protein solution was added to the MALDI target plate, mixed with 0.5 µL matrix solution (5 mg/mL a-Cyano-4-hydroxycinnamic acid in 80% Acetonitrile, 0.1% TFA) and dried in. Intact α-synuclein protein samples were analysed in linear positive mode on a MALDI mass spectrometer (Autoflex Speed, Bruker Daltonics) using external calibration with protein calibration standard I (Bruker Daltonics).

#### 4.10.2. Digestion of Samples for Quantification Using Isotope Standard (^14^N vs. ^15^N)

Ammonium bicarbonate was added to the α-synuclein protein samples to a final concentration of 100 mM. Trypsin was then added to a tryspin:protein ratio 1:50, and the samples were digested overnight at 37 °C. Digestion was stopped by addition of formic acid (FA) to a final concentration of 0.5%.

#### 4.10.3. LC-MSMS

Digested peptide samples were cleaned up on reversed-phase C18 micro-columns before injected to an ultra-high pressure nanoflow chromatography system (nanoElute, Bruker Daltonics). The peptides were loaded onto an Acclaim PepMap C18 (5 mm, 300 μm id, 5 μm particle diameter, 100 Å pore size) trap column (Thermo Fisher Scientific) and separated on a Bruker Pepsep Ten C18 (75 µm × 10 cm, 1.9 µm particle size) analytical column (Bruker Daltonics). Mobile phase A (2% ACN, 0.1% FA) was used with the mobile phase B (0.1% FA in ACN) for 45 min to create a gradient (from 2 to 17% B in 20 min, from 17 to 34% B in 10 min, from 34 to 95% B in 3 min, at 95% B for 12 min) at a flow rate of 400 nl/min and a column oven temperature of 50 °C. The peptides were analysed on a quadrupole time-of-flight mass spectrometer (timsTOF Pro, Bruker Daltonics) via a nano electrospray ion source (Captive Spray Source, Bruker Daltonics) in positive mode, controlled by the OtofControl 5.1 software (Bruker Daltonics). The temperature of the ion transfer capillary was 180 °C. A DDA method was used to select precursor ions for fragmentation with one TIMS-MS scan and 10 PASEF MS/MS scans. The TIMS-MS scan was acquired between 0.60–1.6 V s/cm^2^ and 100–1700 *m*/*z* with a ramp time of 100 ms. The 10 PASEF scans contained a maximum of 10 MS/MS scans per PASEF scan with a collision energy of 10 eV. Precursors with maximum 5 charges with intensity threshold to 5000 a. u. and a dynamic exclusion of 0.4 s were used.

#### 4.10.4. Data Analysis

Raw data were processed using Mascot Distiller (version 2.8) and the quantification was performed using Mascot Quantitation Toolbox with the 15N Metabolic Quantitation. All data were searched against the SwissProt Database using the settings precursor ion tolerance 10 ppm, MS/MS fragment mass tolerance 0.015 Da, trypsin as protease, 1 missed cleavages site. For the quantification, six different α-synuclein peptides were used, all identified with significant individual ion scores (*p* < 0.005) at least two times for each light (L, ^14^N) and heavy (H, ^15^N) version of each peptide searched against the SwissProt Database. First, all six peptides are presented with an L/H value and then an average is calculated based on these to obtain a final L/H value for the protein. This final L/H value was then used to calculate the concentration of the ^15^N α-synuclein in the sample using the known amount of ^14^N α-synuclein that was added to the samples.

#### 4.10.5. Peptides Used for Quantification Using Isotope Standard (^14^N vs. ^15^N)

The following α-synuclein peptides were used to quantify and determine the α-synuclein concentration in the samples: EGVVAAAEK (2+), EGVLYVGSK (2+), AKEGVVAAAEK (2+), TKEGVLYVGSK (2+), EGVVHGVATVAEK (2+), TKEGVVHGVATVAEK (3+).

## Figures and Tables

**Figure 2 ijms-24-05191-f002:**
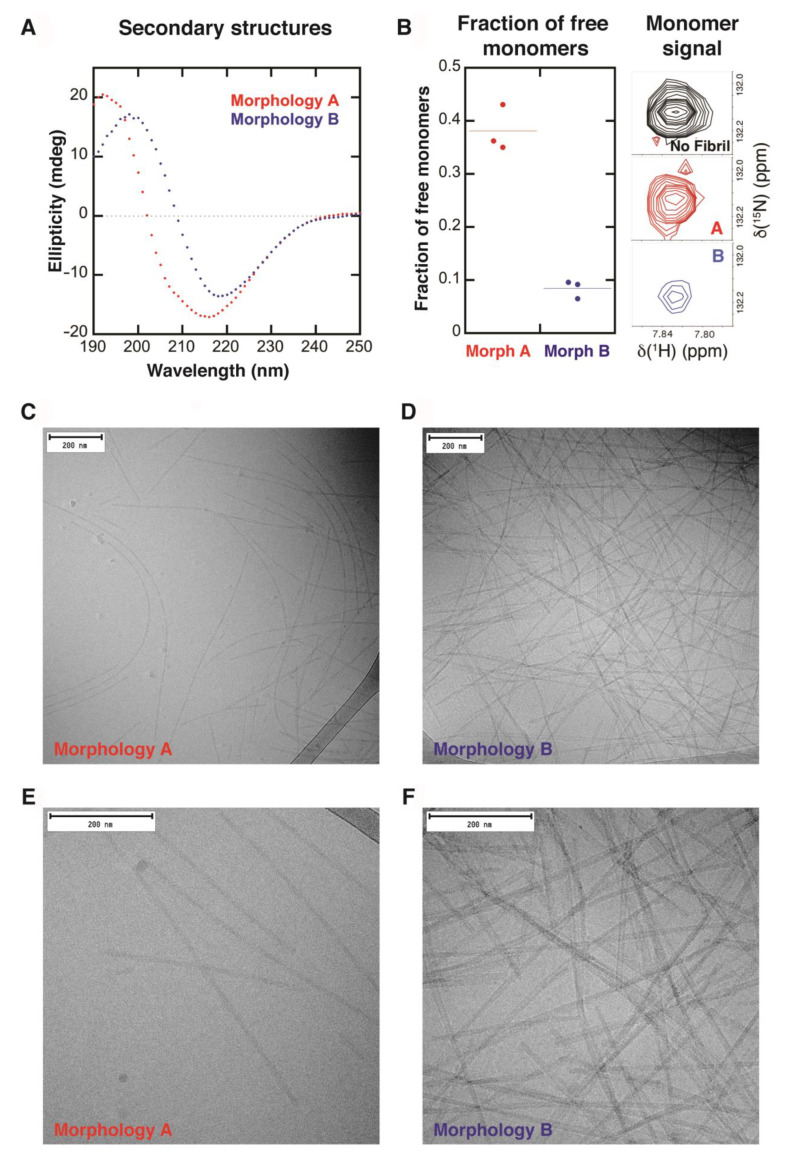
Two different fibril morphologies, A and B, observed in two different samples originating from the same monomer preparation (10 mM Tris, 0.02% NaN_3_, 5% D_2_O, pH 7). (**A**) CD spectra in the secondary structure region for the two different morphologies. (**B**) **Left**: Fraction of free monomers measured by NMR spectroscopy by comparing the relative integrated intensity of A140 in the fibrillar samples relative to that of the monomeric sample. The data are visualized with a dot-plot, where each data point is represented with a dot and the average with a dashed line. **Right**: Comparison of the ^15^N−^1^H signals corresponding to A140 for monomer only, monomer in the presence of morphology A, and monomer in the presence of morphology B. (**C**–**F**) Cryo-TEM images showing difference in ultrastructure (morphologies) between the two fibril samples. Figures (**C**,**D**) show images of 40k magnification and figures (**E**,**F**) show images of 80k magnification.

**Figure 3 ijms-24-05191-f003:**
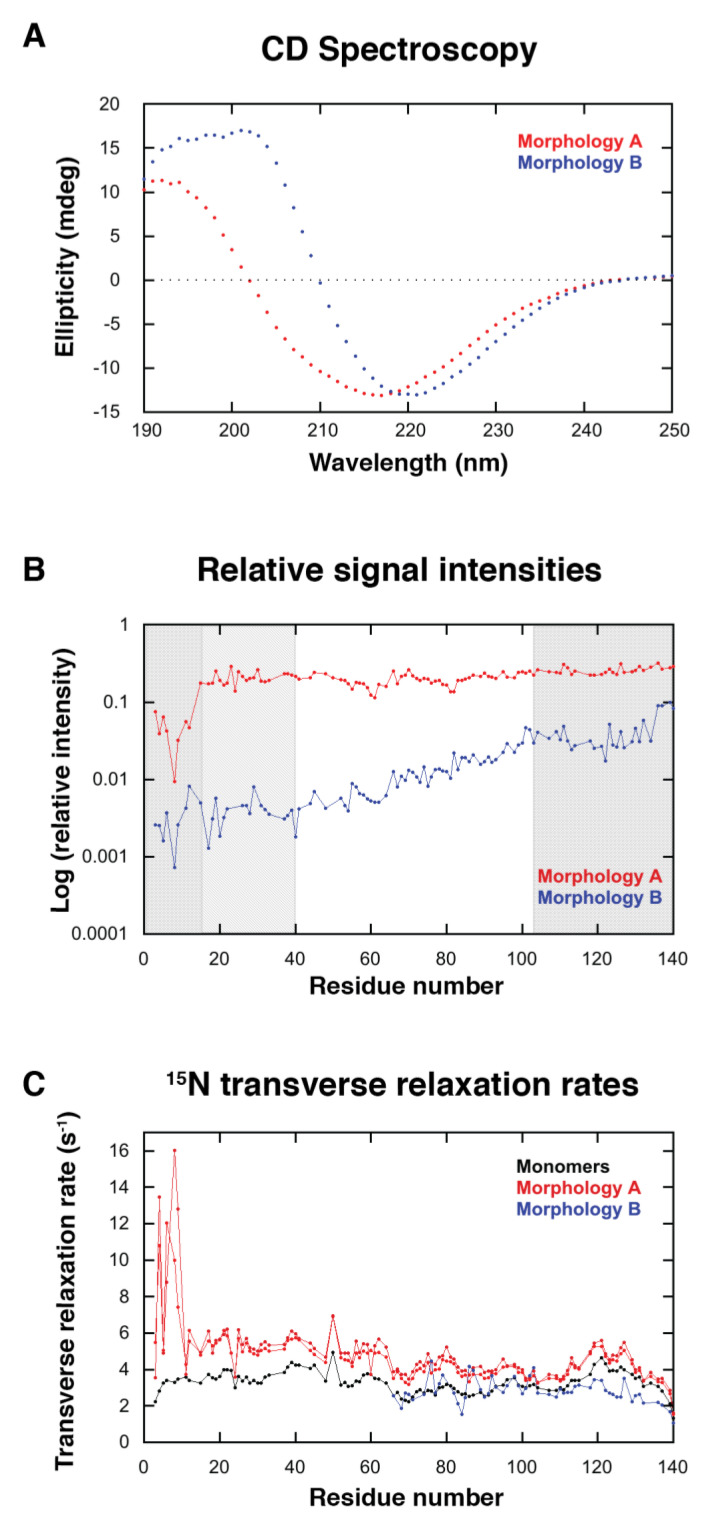
NMR characterization of transient monomer fibril interactions for different morphologies. (**A**) CD spectra of samples containing fibrils of morphology A and morphology B. These samples were prepared at the same conditions (10 mM Tris, pH 7, 0.02% NaN_3_, 5% D_2_O. (**B**) Residue-resolved signal intensities of monomers in the presence of fibril relative to the intensities in the same solution before formation of fibrils. The shaded areas mark residues 1–15, 16–40, and 102–140. (**C**) Residue-resolved ^15^N transverse relaxation rates for monomers in the absence and presence of fibrils. Monomers in the absence of fibrils are in black, monomers in the presence of morphology A are in red, and monomers in the presence of morphology B are in blue. In B, the results from two separate measurements in the presence of morphology A are shown to illustrate the reproducibility of the results. In B, relaxation rates could only be measured from V66 and onwards.

**Figure 4 ijms-24-05191-f004:**
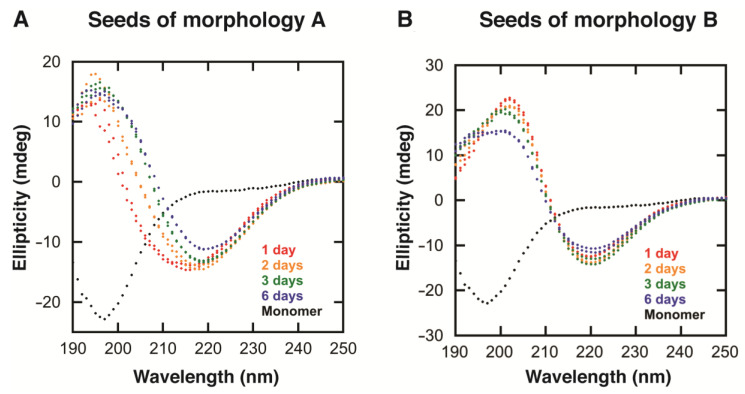
The replication of morphologies (**A**,**B**) using 1% seed. The different graphs show CD spectra of the monomeric sample (black), supplemented with 1% seeds of morphology (**A**) or 1% seeds of morphology (**B**) before incubation at 37 °C with stirring for 6 days. The CD spectra were measured after 1 day (red), 2 days (orange), 3 days (green), and 6 days (blue). Apart from the monomer, two replicates are shown for each time point.

**Figure 5 ijms-24-05191-f005:**
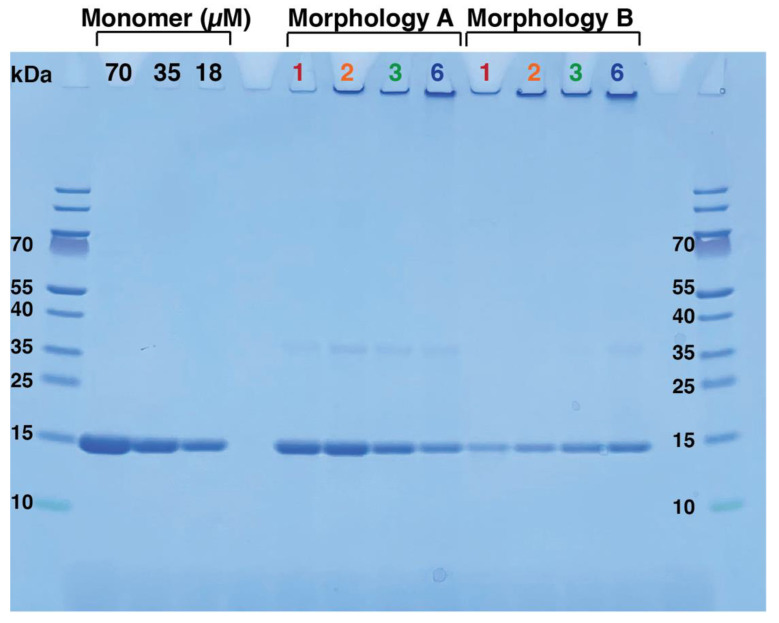
SDS-PAGE containing supernatant of samples seeded with morphology A and morphology B. Aliquots were taken from fresh monomer as well as from seeded sample after incubation for: 1 day; 2 days; 3 days; and 6 days at 37 °C with stirring. Separation of fibrils and monomers was performed as explained in Section 4.8. The first and last well contain Mw standard. For comparison of concentrations, monomeric sample was loaded at three different concentrations, 70 µM, 35 µM and 18 µM (black). Supernatant of samples containing fibrils of morphologies A and B, after 1, 2, 3, and 6 days incubation are shown in red, orange, green and blue, respectively. The quantification of α-synuclein concentration in these samples using isotope standard and MALDI mass spectrometry after digestion with trypsin are shown in Appendix A.

**Figure 6 ijms-24-05191-f006:**
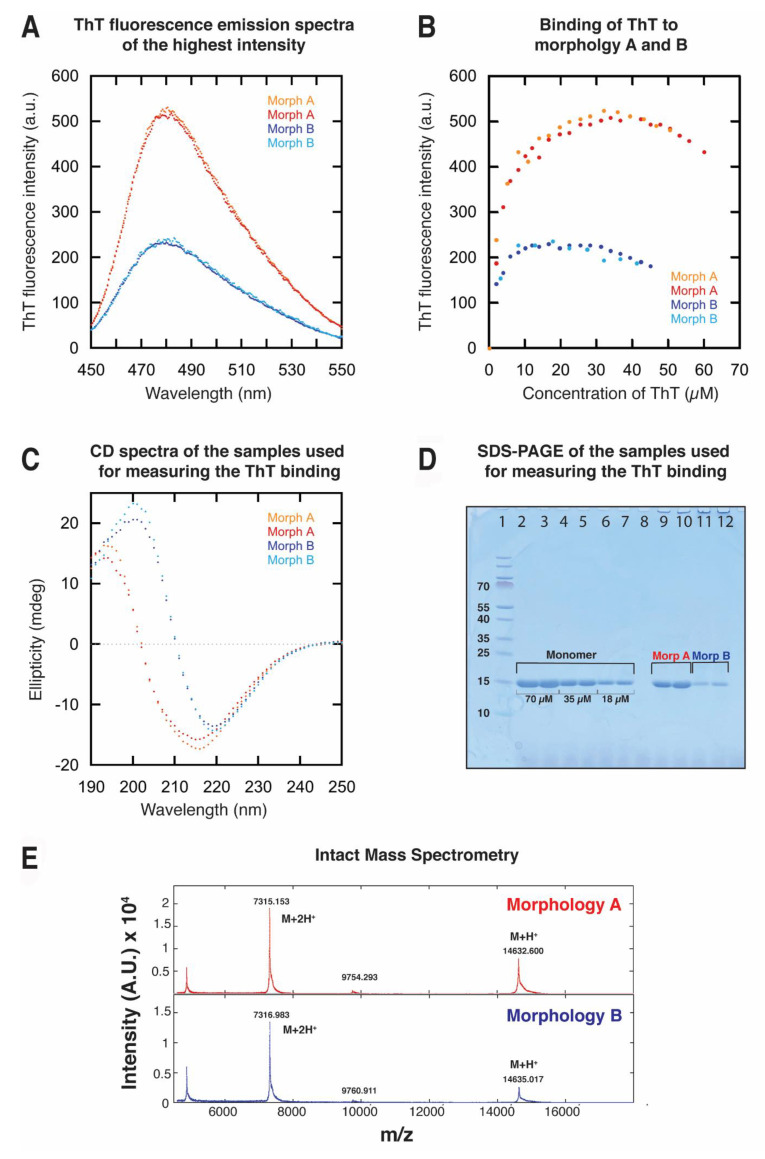
Investigation of surface properties of morphologies A and B by measuring binding of ThT. (**A**) ThT fluorescence emission spectra of morphologies A and B, showing the full emission spectra of the highest intensity. The emission spectra of two individual samples are shown for each morphology: morphology A (orange and red) and morphology B (dark and light blue). For morphology A, the maximum intensity was observed after addition of 36 and 34 µM ThT. For morphology B, the maximum intensity was observed after addition of 17 and 18 µM ThT. The ThT fluorescence intensity for morphology A was more than double the intensity of morphology B: 531 and 518 for the two replicates of morphology A, and 233 and 243 for the two replicates of morphology B. (**B**) Binding of ThT to morphologies A and B. The graph shows the maximum fluorescence intensity (average intensity between 477 and 482 nm) plotted against concentration of ThT. The maximum fluorescence intensity was reached at lower concentration of ThT for morphology B than A. Titration of ThT into two individual samples are shown for each morphology. (**C**) CD spectra of the samples used for these experiments. (**D**) An aliquot of each sample used for the experiment was centrifuged and the supernatant was collected and loaded onto an SDS-PAGE in order to compare the concentration of soluble monomers present in each sample. The fibrillar samples were made from 70 µM monomeric sample (well 2–3). Wells 3–6 show a serial dilution (70, 35, and 18 µM) of the monomeric sample, used for comparison with the concentration of monomers present in the supernatant. Each sample was loaded in duplicate. The supernatant collected from the two individual samples of each morphology are shown in wells 9–10 (morphology A) and 11–12 (morphology B). (**E**) MALDI mass spectra of α-synuclein monomers in the supernatant over fibrils of morphology A (**top**) and morphology B (**bottom**).

**Figure 7 ijms-24-05191-f007:**
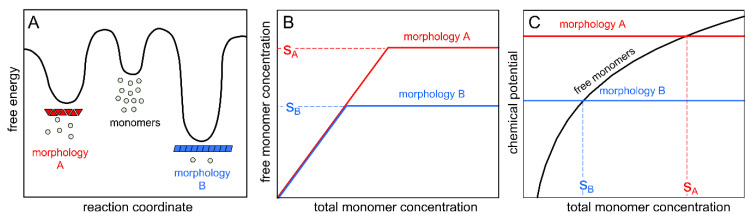
Schematic representation of the energy landscape and the solubility of morphologies A and B. (**A**) Simplified representation of the energy landscape for morphologies A and B, showing that morphology B (associated with a lower apparent solubility) is thermodynamically more stable (lower free energy). Primary nucleation is a very rare event. Once a seed of morphology A is formed in a sample, the replication of this morphology by elongation and secondary nucleation is energetically favored over primary nucleation, thus initially suppressing the emergence of the more stable morphology B. If a seed of morphology B is formed in a sample, the replication of this morphology by elongation and secondary nucleation is energetically favored over primary nucleation, thus suppressing the emergence of the less stable morphology A over time. Thus, the high effective energy barriers between morphologies A and B, and the pathway via monomers and de novo nucleation of fibrils of morphology B, may explain the kinetic stability of morphology A. (**B**) A simplified representation of the lower apparent solubility of the thermodynamically more stable morphology B (S_B_) in comparison to that of morphology A (S_A_). (**C**) Schematic illustration of the concentration-dependent chemical potential of free monomers (black curve) and the concentration-independent chemical potentials of monomers in fibrils of morphologies A (red line) and B (blue line). The crossover points correspond to the apparent solubility of each morphology.

**Figure 8 ijms-24-05191-f008:**
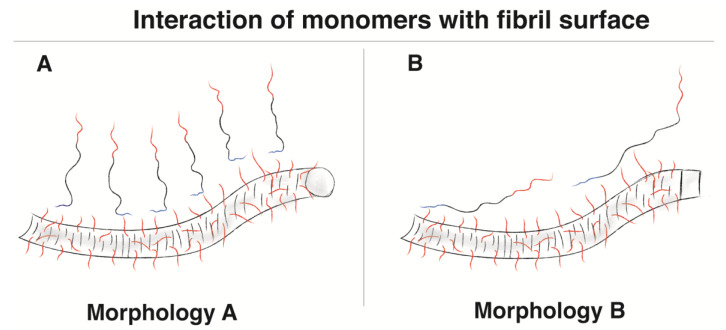
Simplified illustration comparing the interactions of monomers with the surface of morphologies A and B. (**A**) Monomers interacting with the surface of morphology A, showing how mainly the first residues of the N-terminus interact with the fibril surface. (**B**) Monomers interacting with the surface of morphology B, showing how a large part of the monomer is affected by the interaction between the monomers and the fibril surface, where the N-terminus and probably also the N-terminal part of the NAC region is the main binding site but that the rest of the molecule also interacts to some degree.

## Data Availability

All data will be made available by one of the corresponding authors upon reasonable request.

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
