# Peer review of "Morphology-Dependent Interactions between α-Synuclein Monomers and Fibrils"

_ijms, 2023, doi:10.3390/ijms24065191_

Round 1

Reviewer 1 Report

Reviewer #1: The paper characterized two different morphologies of a-synuclein fibrils under identical solution conditions and the protein sequence, revealed by NMR, CD, and fluorescence spectroscopy, as well as by Cryo-TEM. The authors showed that morphology A and B have different structures, surface properties, binding properties, and solubility, importantly, morphology B is more thermodynamically stable than morphology A.  The data are high-quality, and the manuscript is well-written. The discussion in the manuscript is sound. Therefore, the manuscript is suitable for publication in the International Journal of Molecular Sciences after minor revisions, as suggested below.

 1. Figure 1a should be referred to in the text somewhere.

2. Line 876, the reference format should be consistent with others, remove the DOI number.

Author Response

  1. Figure 1a should be referred to in the text somewhere.

Response: This is now cited on page 2, line 56.

  1. Line 876, the reference format should be consistent with others, remove the DOI number.

Response: Done.

Reviewer 2 Report

The interesting paper by Palmadottir et al presents the results on the identification and characterization of two morphologically distinct forms of a-syn fibrils formed in different samples under identical in vitro conditions.

While it is known that a-syn can form morphologically distinct fibrils under different experimental conditions, the results presented in the manuscript by Palmadottir et al are interesting because these two fibrillar states, originating from the same starting a-syn monomers, are thermodynamically different and exchange over time toward the most stable form.

A variety of experimental methods have been employed to characterize the two morphs and also to investigate possible interaction modes of a-syn monomers with both fibrillar states. The experiments are well described, and appropriate control experiments have been carried out.

Minor points

-       Legend to Fig. 6: the text related to Panel D describes the SDS-PAGE but in the Figure Panel D shows the results of Mass Spec. (Panel E) which is not described in the legend.

-       Page 3 line 115: (Figure 1 b and c)

-       Page 21 line 712 (C)

Author Response

-       Legend to Fig. 6: the text related to Panel D describes the SDS-PAGE but in the Figure Panel D shows the results of Mass Spec. (Panel E) which is not described in the legend.

Response: This is now corrected in the legend and text.

-       Page 3 line 115: (Figure 1 b and c)

Response: This is now corrected.

-       Page 21 line 712 (C)

Response: This is now in bold.

Reviewer 3 Report

The authors expressed recombinant alpha-synuclein and observed the formation of two different fibril types under the same condition using multiple techniques, including NMR, CD, fluorescence spectroscopy and cryoTEM. The as the fibrils form, fibrils forming of the A type show more monomers in solution while fibrils forming of the B type show more polymerization and less monomers in solution. The authors show that over time fibrils of the A type will convert to the B type. This is a really nicely presented, well written, manuscript with results from multiple techniques supporting the authors' conclusions.

minor concerns:

Figure 2b and monomer amounts, perhaps I missed it but the authors should state whether they are using peak intensity or integrated volume to gauge the amount of monomer signal.

Figure 5 - morphology lanes A have a second band around double the bands (30kDa) of the alpha-synuclein monomer (15kDa) that is not present in the morphology B lanes or the monomer lanes. The authors should speculate as to what is going on here as the have very pure protein. The intensity of the about 30kDa band appears highest in the 2 and decreases on day 3 and 6 presumably as the morphology A fibrils start and continue to change to the B type. Also, you see the monomers at the 15kDa decrease in intensity as they are presumably incorporated into B. This deserves some comment and speculation by the authors as to what might be going on. Perhaps this should suggest some future experiments as well.

Author Response

Figure 2b and monomer amounts, perhaps I missed it but the authors should state whether they are using peak intensity or integrated volume to gauge the amount of monomer signal.

Response: Integrated intensity was used as now stated on line 470, page 13.

Figure 5 - morphology lanes A have a second band around double the bands (30kDa) of the alpha-synuclein monomer (15kDa) that is not present in the morphology B lanes or the monomer lanes. The authors should speculate as to what is going on here as the have very pure protein. The intensity of the about 30kDa band appears highest in the 2 and decreases on day 3 and 6 presumably as the morphology A fibrils start and continue to change to the B type. Also, you see the monomers at the 15kDa decrease in intensity as they are presumably incorporated into B. This deserves some comment and speculation by the authors as to what might be going on. Perhaps this should suggest some future experiments as well.

Response: We have added one sentence on line 589-91 in the paragraph regarding the change in monomer concentration over time: “Fibrils of morphology A are thus associated with a higher apparent solubility than fibrils of morphology B”.

We have also added in line 596-600 regarding the dimer band that appears at 30 kda:

” The intensity of the band at 30 kDa is largely proportional to the band at 15 kDa, and thus more visible at the higher monomer concentrations in the presence of morphology A. The oligomeric distribution in the presence of morphology A and B may be further investigated by methods such as Photo-Induced Cross-linking of Unmodified Proteins (PICUP).”